# Influence of Surface Properties and Microbial Growth Media on Antibacterial Action of ZnO

**Dustin Johnson** [1,*] **, John M. Reeks** [2] **, Alexander Caron** [3] **, Iakovos Tzoka** [4] **, Iman Ali** [3] **, Shauna M. McGillivray** [3] **and Yuri M. Strzhemechny** [1]

1   Department of Physics & Astronomy, Texas Christian University, Fort Worth, TX 76129, USA
2   Institute of Low Temperature and Structure Research, Polish Academy of Sciences, 50-422 Wrocław, Poland
3   Department of Biology, Texas Christian University, Fort Worth, TX 76129, USA
4   Department of Physics, University of Texas at Arlington, Arlington, TX 76019, USA
*   Correspondence: dustin.johnson@tcu.edu

**Abstract:** Nano- and microscale ZnO demonstrate robust antibacterial action, although the driving mechanisms remain undetermined. In this study for commercial ZnO nano-powders and home-grown ZnO microparticles of varying morphologies we probe the response to bacterial growth media in isolation and with *Staphylococcus aureus* bacteria. ZnO microparticles are synthesized via a controllable hydrothermal method and subjected to biological assays with varying microbial environments. Changes in the optoelectronic, structural and chemical properties of these crystals before and after such exposure are characterized utilizing temperature-dependent photoluminescence spectroscopy, scanning electron microscopy and energy-dispersive X-ray spectroscopy. This is done to evaluate the impact of surface-surface interactions in antibacterial assays and the role ZnO surface and morphological properties play in these processes. In our experiments various bacterial environments are employed to elucidate the effects of media interactions on the cytotoxic efficacy of ZnO. In particular, minimum inhibitory concentration assays with *Staphylococcus aureus* reveal that microscale particles exhibit antibacterial efficacy comparable to that of the nano-powders, indicating that intra-bacterial internalization is not necessary for antimicrobial action. In our studies we determine that the nature of structural and optoelectronic changes in ZnO depends on both the media type and the presence (or absence) of bacteria in these media. Further evidence is provided to support significant cytotoxicity in the absence of particle internalization in bacteria, further highlighting the role of surface and media interactions in this process.

**Keywords:** ZnO; antibacterial; bacterial growth media; *Staphylococcus aureus*; surfaces





## 1. Introduction

Bacterial infections pose an increasing threat to global health and food security as antibiotic resistant strains become both more prevalent and difficult to treat. The issues associated with the treatment of such resistant strains using traditional antibiotics stem from powerful bacterial adaptive responses in the form of genetic mutations and lateral gene transfer. This makes the development of novel antibiotics increasingly difficult and less profitable, thus necessitating the search for non-traditional antibacterial agents for usage in sanitation and medicines. Such agents should exhibit reduced toxicity, increased selectivity and efficacy against existing antibiotic-resistant strains. This can be achieved through simultaneously implementing multiple modes of action or bypassing resistance mechanisms through activation of alternative pathways from traditional antibiotics. Effectiveness against a wide range of both Gram-positive and Gram-negative bacteria as well as inhibition of biofilm formation would greatly expand the range of microbial species in which growth inhibition could be attained. Additionally, it is desirable to employ materials which are inexpensive, readily available, and whose synthesis is both simple and scalable. In this regard inorganic nanoparticles (NPs) are particularly promising. Size reduction

effects in NPs lead to new mechanical, chemical, and optoelectronic properties as the increased surface-to-volume ratio magnifies the substantially increased contribution of surface characteristics. The latter, in turn, are interesting from an antibacterial perspective as various nanomaterials have demonstrated bactericidal efficacy in vitro, in vivo, and in animal models [1,2]. The scale factor alone provides a massive active and dynamic interface between the NPs and the bacterial cell wall, leading them to be critical components of various antibacterial coatings [3,4]. Many inorganic NPs have the additional benefit of high biocompatibility and possess novel antibacterial modalities on their own, thus further adding to their utility as an alternative to organic antibiotics.

Nanomaterials currently play a key role in such industries as healthcare, water treatment, textiles, food storage and transportation due to their antibacterial properties [3–6]. A particularly promising set of compounds are those based on ZnO [2–4,7,8]. A wide range of ZnO compounds demonstrate notable antibacterial action against antibiotic resistant strains of Gram-positive and Gram-negative bacteria under various conditions with selective toxicity [9–11] and effectiveness in combating biofilm formation. ZnO is an abundant, low-cost material with significant photocatalytic efficiency and high biocompatibility, being recognized as a safe substance for exterior applications by the US Food and Drug Administration [12,13]. Additionally, ZnO is a stable compound of high durability and heat resistance leading to widespread usage in antibacterial coatings for food storage, biomedicine, metallurgy, and chemical industries [3,4,9,10,12,14–16]. Despite ubiquitous applications and abundant scientific attention, there exists a significant debate about the fundamental mechanisms behind the observed antibacterial behavior. Such gaps in understanding limit the scope and effectiveness of current bactericidal applications and hinder the development of new ones. Much work has been done to isolate the driving antimicrobial mechanisms by focusing on the bacterial response in biological assays with NPs [17–20]. Presently, the prevailing theories as to the nature of these interactions point to a number of mechanisms.

The most commonly referenced mechanism is generation of reactive oxygen species (ROS) [17,19]. ROS disrupt bacterial function via internalization and subsequent destruction of vital cellular components such as DNA, proteins and lipids. In ZnO, the predominant explanation for the ROS production is photocatalytic generation mediated by ultraviolet (UV) light. In this process excited electrons dissociate water molecules allowing for dissolved $O_2$ to react with free ions and form potent ROS such as $H_2O_2$. This explanation, however, is incomplete since the observation of notable antibacterial activity occurs even in the absence of UV radiation [20–22]. Few alternative explanations have been suggested. In particular, some authors proposed that ROS generation may be linked to superoxide anion production at the ZnO surface [23]. Such explanations require further investigation and raise questions as to the role of the free crystalline surface in this process. At present, ROS generation mechanisms in ZnO are not well understood leaving many questions about generation pathways and the significance of such processes in the antibacterial action as such.

Another often cited mechanism is that of a cellular dysfunction stemming from the cation toxicity [17,24,25]. $Zn^{2+}$ ions possess biomimetic properties with well-established pathways for internalization. $Zn^{2+}$ can appropriate the $Fe^{2+}$ ion channels and subsequently replace $Fe^{2+}$ during DNA transcription. This causes inability to repair the cellular membrane leading to cell dysfunction and death. However, this explanation is challenged by the general insolubility of crystalline ZnO in water with only partial solubility observed in biologically relevant media [26]. Additionally, attempts to quantify the free ion activity demonstrated that inhibition is weak and generation of zinc ions is too low to explain the observed level of cytotoxicity [27,28].

Additionally, as stated in the literature, cell internalization of ZnO NPs can result in a disruption of cellular processes through physical contact or generation of harmful chemical species such as ROS and $Zn^{2+}$ ions. This determination is supported by a clear relationship between particle size and antibacterial activity for ZnO NPs [17,20,26]. Yet, it is unclear if

internalization of ZnO NPs leads to degradation of the cell wall or if external disruptions of the cell wall allow for the observed internalization. Studies into this phenomenon indicate that the latter may be more likely. Previous work has shown that for *Staphylococcus aureus* (*S. aureus*), ZnO microparticles (MPs) far too large to be internalized by the bacteria, demonstrated cytotoxic efficacy comparable to those at the nanoscale [29]. These studies indicate that cell internalization of ZnO is not a driving force in the observed antibacterial behavior, pointing to other mechanisms that scale with particle size [29,30].

One such mechanism could be associated with interactions between growth media, bacteria and ZnO, as they all relate to antibacterial action. In particular, interactions with the crystalline free surface might be vital [18,20,21]. Such surface specific mechanisms are difficult to isolate through bacterial studies as ZnO possesses an anisotropic wurtzite lattice structure, resulting in two distinctive surface types. These are either polar hexagonal faces or relatively non-polar rectangular faces. These surface types possess differing electrochemical properties that may cause them to participate in antibacterial interactions preferentially/differently. As of today, studies in this area are rather scarce, nevertheless some promising results have been obtained in which antibacterial activity was linked to general abundance of surface oxygen vacancies affecting charge dynamics at the bacterial cell wall [20]. Such findings indicate that there remains a need to evaluate both the overarching role of the free surface in the antibacterial action of ZnO and if/how these interactions differ between prevalent surface types.

While all the aforementioned mechanisms are plausible and prevalent within the literature, conflicting evidence has led to vigorous debate. It should be noted that the non-specificity of particle activity may mean that these behaviors are explained by more than one mechanism, further complicating the task of interpreting the underlying behavior. The interdependence and complexity of these interactions highlight the importance of studying not just the response of the bacteria being acted upon but also that of the ZnO particle surfaces. To address this matter, we shift the focus towards the response of the antibacterial agent to microbial environments. This is a novel approach since previous studies in the field concentrated primarily on the response of bacteria or the corresponding bacterial environments. In our work, we investigate the influence of the ZnO free surfaces in antibacterial interactions and changes therein. These investigations differentiate themselves from the current landscape of research in this area as we utilize microscale particles to both eliminate internalization and to better elucidate the interactions at the free crystalline surface with a specific polarity. We focus on not just the influence of growth media on cytotoxicity but on isolating those interactions between just ZnO and growth media as well as those with bacteria.

As such, we utilize temperature-dependent photoluminescence (PL) spectroscopy, field emission scanning electron microscopy (FE-SEM) and energy-dispersive X-ray spectroscopy (EDXS) to study changes in the optoelectronic, structural and chemical properties of varying morphologies of commercially acquired and home grown ZnO NPs and MPs, the latter being used to control for effects related to internalization. These studies are performed before and after antibacterial assays with *S. aureus* in Mueller Hinton Broth (MHB), saline, and phosphate-buffered saline (PBS) growth media solutions. This is done to elucidate what modifications of ZnO characteristics are responsible for bacterial growth inhibition and evaluate the role of interactions between ZnO and bacterial growth media. Identical investigations are performed with ZnO particles exposed to growth media alone (without bacteria) to probe the nature of interactions between ZnO and bacterial growth media. Herein we find that biological assays and structural changes in ZnO confirm that the microbial environment plays a substantial role in both the nature of and capacity for bacterial growth inhibition of ZnO. Optical studies point to the differing nature of interactions between bacteria and those of varying growth media. We also demonstrate a reduced dependency on internalization of ZnO particles than previously reported. These results described below, are indicative of the existence of competitive processes between aqueous phosphates in the bacterial growth environment and *S. aureus* bacteria that influence the

driving mechanisms of antibacterial action. We demonstrate interactions with optically active defects associated with the production of excess $Zn^{2+}$. The changes resulting from these interactions are shown to occur due to either direct interactions of *S. aureus* with ZnO surface oxygen deficiencies or the increased solubility of associated surfaces leading to excess $Zn^{2+}$ in the bacterial environment.

## 2. Materials and Methods

### 2.1. ZnO Synthesis

Commercial ZnO NPs (with the average size < 100 nm) were provided by Sigma Aldrich (SA, St. Louis, MO, USA) and Zochem Inc (Dickson, TN, USA). MPs were synthesized in-house using a bottom-up hydrothermal growth method previously described in [29] as follows. ZnO microcrystals were grown from a D.I. water, a Zn salt in the form of 1 M $Zn(CH_3CO_2)_2 \cdot H_2O$ and 1M $(CH_2)_6N_4$ with 99.999% pure Zn foil supplied by Sigma Aldrich as an additional Zn source. This solution was then allowed to react in an autoclave at temperatures in excess of 90 °C to catalyze the reaction. After which, the resulting solution underwent centrifugation and removal of the supernatant. The remaining solid was then washed and dried with D.I. water and acetone. Adjustment of such parameters as relative precursor concentrations, temperature, reaction time, pressure and pH allowed for synthesis of ZnO MPs with a high quality of the crystalline free surface, tunable size/morphology and a well-controlled relative abundance of polar and non-polar surfaces.

### 2.2. Bulk Physical and Chemical Characterization

Confirmation of the predominant surface morphology and chemical composition of ZnO crystals was done by the SEM and EDXS techniques, utilizing a JEOL FE-SEM instrument (JEOL, Peabody, MA, USA) at an operating voltage of 15 kV and a probe current of 9.6 A. Samples were mounted onto a carbon tape on an aluminum mount before being placed into the chamber. Alternate preparation was made for samples exposed to bacteria. To prepare these for FE-SEM imaging, the remaining cultures from bacterial assays were washed and then fixed in 1.6% glutaraldehyde for 1 h at room temperature. The samples were dehydrated in a series of 10-min ethanol incubations at the following concentrations: 30%, 50%, 70%, 85%, 90%, and two times at 100%. The samples were then incubated with hexamethyldisilazane overnight with the lid open until the excess liquid evaporated. The dry pellets were crushed, transferred to a metal pedestal and sputter-coated with 8 nm of gold in order to produce images of the bacteria. Surface area calculations and polarity abundances were determined employing the ImageJ software (version 1.51) on the SEM images. These tools were used before and after bacterial assays with ZnO incubated with *S. aureus* in MHB, saline, or PBS.

### 2.3. Antimicrobial Assays

The baseline bactericidal potential in our samples was established by performing MIC assays on *S. aureus* Newman strain, a methicillin-susceptible *S. aureus* strain. These assays were performed in 3 different mediums; MHB (Fisher Scientific), saline, or PBS. Saline and PBS were prepared by dissolving 4.0 g NaCl, 0.1 g KCl (saline) or 4.0 g NaCl, 0.1 g KCl, and 0.72 g anhydrous sodium phosphate-dibasic (PBS) in 500 mL of water, adjusting the pH to 7.4 and autoclaving. *S. aureus* was grown to early log phase (OD600 of 0.4) in MHB, washed, and resuspended in MHB, saline or PBS and then diluted 1:100 in the respective medias. Diluted *S. aureus* cultures were incubated at a 1:1 ratio with 5 mg/mL of ZnO suspended in the same media-type for a final concentration of 2.5 mg/mL ZnO and 1:200 dilution of log-phase *S. aureus* in 1.5 mL. Tubes were mixed by rocking or inverting at 37 °C and 200 µL of culture from each tube was removed at the indicated time points, centrifuged at 100 rcf for 2 min to pellet the ZnO particles, and serial dilutions of the supernatant were plated to enumerate surviving CFU/mL. Larger volumes of ZnO samples used for optoelectronic characterization used the same ratios as the survival assays but were scaled up to a larger volume ($2 \times 50$ mL tubes per condition).

### 2.4. Optoelectronic Characterization

PL spectra were taken at temperatures ranging from 10 K to 300 K. ZnO powders were pressed and mounted inside of an evacuated Janis CCS150 cryostat. Excitation at 325 nm was achieved using a Kimmon IK HeCd continuous wave (CW) laser and an accompanying optical train. The resulting luminescence spectra were collected by a Horiba Jobin Yvon T64000 Triple Raman Spectrometer (Horiba, Piscataway, NJ, USA) with a Synapse CCD. These optical studies were performed on the ZnO samples before and after their exposure to the bacterial growth media both with and without the presence of *S. aureus*.

### 3. Results & Discussion

As previously reported [29], we employed the hydrothermal growth technique to produce ZnO MPs of tunable size and morphology to probe the influence of ZnO surface polarity on the underlying antibacterial behavior of ZnO at the micro and nanoscale. The synthesized MPs, in conjunction with commercial NPs, were then used in antibacterial assays and underwent optoelectronic characterization. Samples were characterized via FE-SEM, EDXS and temperature-dependent PL to confirm size, crystal quality, surface polarity, composition and optoelectronic properties. MIC and other bacterial exposure assays were utilized to evaluate the antibacterial efficacy of different morphologies. Characterization was performed before and after these assays to elucidate how such exposures impact morphological and optoelectronic properties of ZnO.

FE-SEM was used to determine morphology and particle size distributions in the investigated specimens, with some results shown in Figure 1. Commercial ZnO samples (Figure 1a) have a relatively random distribution of NP morphologies, whereas the hydrothermally grown MPs span a range of rather well-defined morphologies. Figure 1b depicts ZnO MPs with a characteristic hexagonal prism structure and a relative balance of surface area between polar and non-polar surfaces. Figure 1c shows an example of MPs with an elongation along the *c*-axis (rod-like structures) and hence an increased relative abundance of non-polar faces. The opposite case is shown in Figure 1d where the radial growth of MPs has outpaced the growth along the *c*-axis resulting in flatter plate-like structures with predominantly polar free surfaces. These results demonstrate our ability to produce crystals of tunable morphology with high quality of crystalline surfaces and edges.

In characterizing the MP samples, we sought to classify them according to their relative abundances of polar to non-polar surface types (P/N ratio). For this, ImageJ surface area analysis was performed, the results of which are shown in Table 1. We classify those with a P/N ratio of >1.2 as hexagonal plates (HP), those with a P/N ratio of <0.6 as long rods (LR) and those with intermediate values as balanced (B). We will use these classifications to refer to the morphologies of the hydrothermally grown MP samples throughout this text.

**Table 1.** Ratio of polar and non-polar surface areas for selected hydrothermally grown samples.

| Morphology | Hexagonal Plates | Balanced | Long Rods |
|---|---|---|---|
| **P/N ratio** | 4.22; 3.426 | 0.845; 0.677 | 0.475; 0.255 |

Electronic structure of ZnO NPs and MPs was studied by PL spectroscopy in a wide range of temperatures and photon energies. E.g., Figure 2 juxtaposes luminescent properties of ZnO NPs and MPs in the near-band edge (NBE) region and in the temperature range from 10 K to 300 K. Nanoscale ZnO (Figure 2a) exhibits highly structured spectra at lower temperatures. Here, along with the free excitonic (FEx) emission (at ~3.37 eV for 10 K) one can see at lower energies well-defined bound exciton (BEx) peaks and their associated phonon replicas. As the temperature increases, BEx luminescence gradually dissociates and the broad FEx band dominates. This is in contrast with the NBE spectra shown in Figure 2b for ZnO MPs. Here, we see a suppression of the FEx luminescence in conjunction with significant broadening of the BEx spectral feature. This behavior could be associated with a high concentration of surface-bound excitons as opposed to those bound to bulk point

defect sites. Excitons bound to surface trap states exhibit broader spectral lines due to the decreased localization [31]. Such observation points to possibly a greater abundance of surface trap states in hydrothermally grown ZnO MPs compared to their commercial NP counterparts, which in turn could affect interactions between bacteria and ZnO crystalline surfaces.

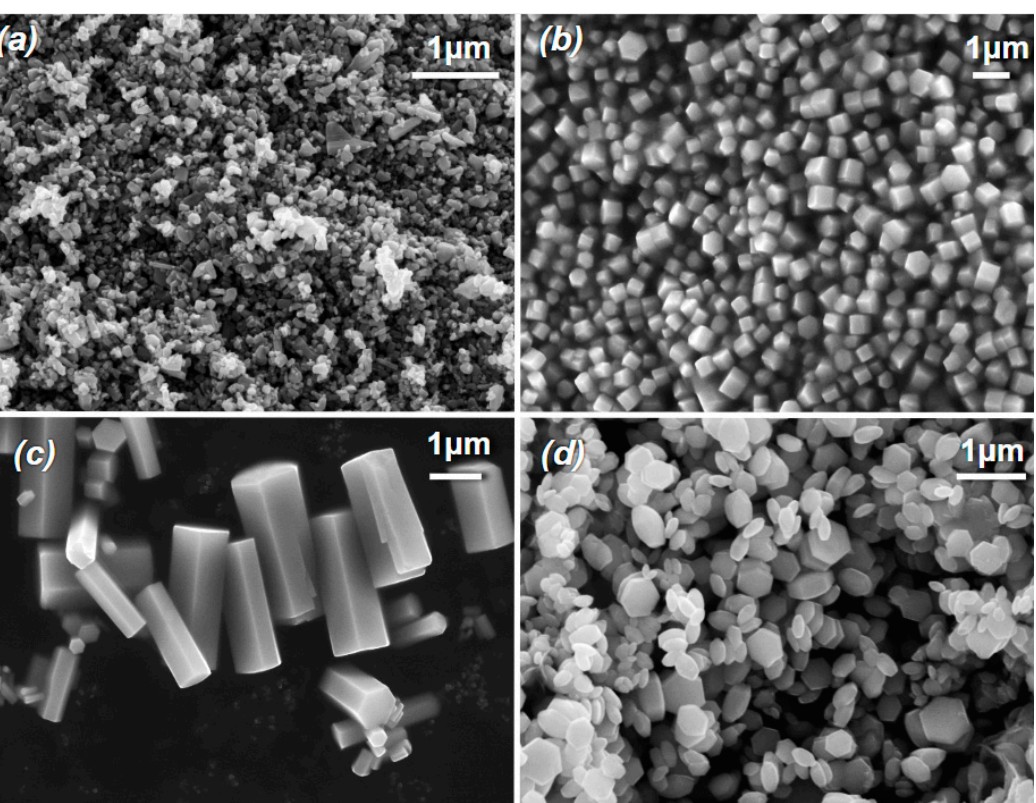

**Figure 1.** FE-SEM images of different sample morphologies: (**a**) commercial ZnO NPs; (**b**) prism-like MPs with a relative balance between polar and nonpolar free surfaces; (**c**) rod-like MPs with more nonpolar surfaces exposed; (**d**) plate-like MPs with more polar surfaces exposed.

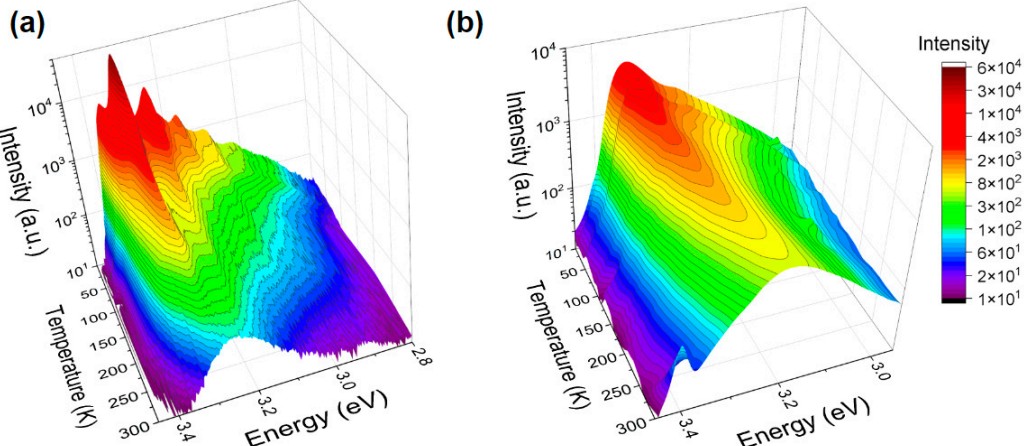

**Figure 2.** Temperature dependencies of PL spectra for: (**a**) as-received commercial ZnO NPs and (**b**) as-grown ZnO MPs.

Furthermore, we observed that the NBE luminescence changes with the MP sample morphology. Figure 3 illustrates this trend in the room temperature (RT) NBE emission. One of the explanations of this behavior could be associated with changes in the crystal field

or the orbital symmetry of the vacant $Zn^{2+}$ $4s$ and the occupied $O^{2-}$ $2p$ states as we change scales. Depending on the relative abundance of polar vs. non-polar surfaces in ZnO MPs, such a change can be attributed to either differences in the charge carrier concentrations at and near the surface or the density of occupied surface trap states. It has been reported [32] that an excess of surface trap states at polar surfaces influences the core level binding of the Zn $2p$ orbital energy, shifting it toward higher values. In particular, this has been demonstrated for the surface states related to oxygen vacancies [32,33]. Alternatively, such trap states may preferentially attract passivating compounds from the ambient during growth. These adsorbates would act as acceptor states depleting surface states, which results in surface band bending [34,35].

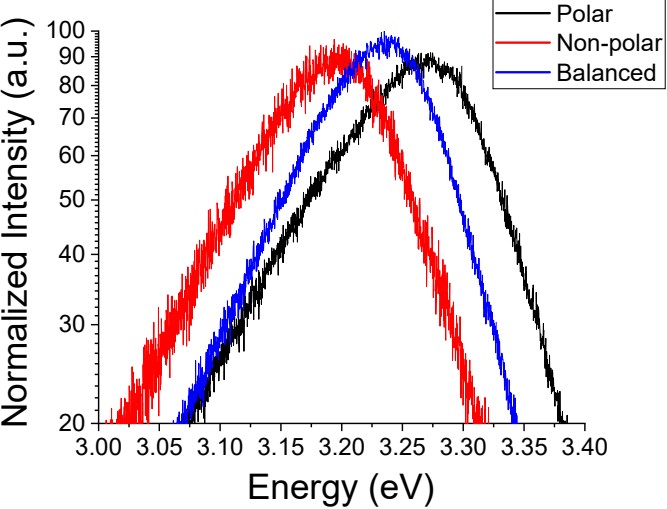

**Figure 3.** Room temperature PL spectra, normalized to the peak NBE emission, of the NBE region for as-grown ZnO MPs with different morphologies.

In addition, comparing the deep defect luminescence of samples with primarily polar to primarily non-polar morphologies (Figure 4), one can observe a comparable intensity of an emission band centered around ~2.1 eV (Figure 4a), attributed by many authors to Zn vacancies [36–38]. These defects do not appear to be dependent on the predominant surface polarity, but rather on the crystal nucleation and growth conditions.

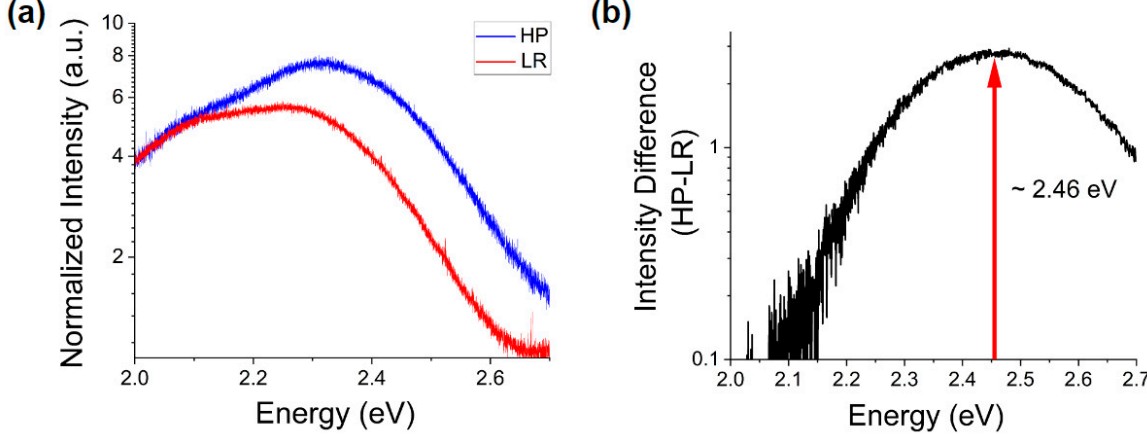

**Figure 4.** (**a**) PL spectra of deep defect emission normalized to the noise floor for as-grown ZnO MPs of differing morphology at 300 K; (**b**) intensity difference of deep defect emission for spectra shown in part (**a**).

Significant differences appear at higher energies, especially when subtracting the two spectra (Figure 4b). One can see for the more polar morphology a pronounced increase of the intensity of the emission band centered at ~2.46 eV. This peak is commonly attributed to oxygen deficiency [39]. This observation is consistent with the observed shifts in the NBE peak energy shown in Figure 3, whence the presence of such deficiency and the previously noted abundance of surface trap states would suggest an excess of $Zn^{2+}$ ions at the polar surfaces of ZnO MPs. Thus, this is an indication of chemical and electronic distinction of the polar surfaces from their non-polar counterparts.

All of the samples studied here were exposed to *S. aureus* bacteria. OD600 results from MIC assays with ZnO NPs are depicted in Figure 5. We find that nanoscale particles exhibit an MIC at ca. 2.5 mg/mL, which is comparable to the MIC previously reported for our hydrothermally grown ZnO microparticles [29]. It should be noted that those results do not indicate differences detectable at this stage in antibacterial action for samples of different synthesis origins and morphological nature.

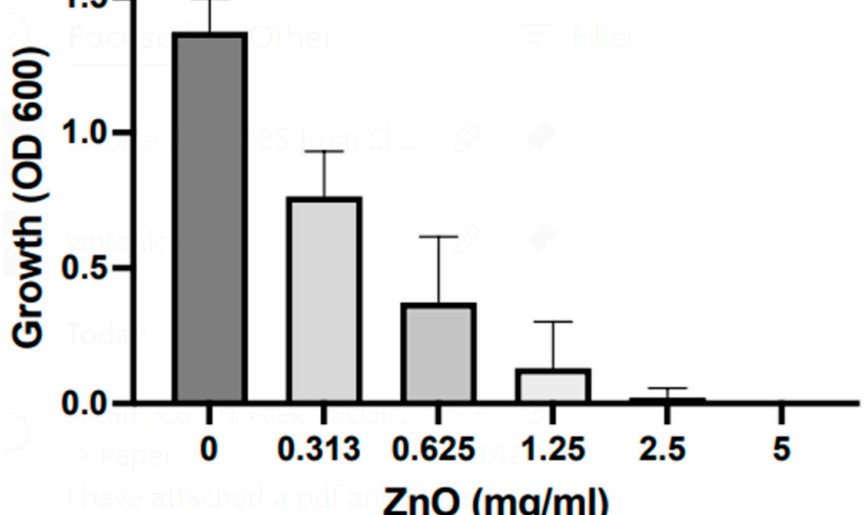

**Figure 5.** MIC results for Sigma Aldrich ZnO NPs with the threshold at ca. 2.5 mg/mL.

This result appears contradictory as the current literature reports on the increasing growth inhibition with the decrease of particle size [4,40,41]. This contradiction could be explained by the fact that in our MIC studies we utilized a novel approach of continuous inversion to maintain contact between bacteria and ZnO particles. Moreover, it is possible that there exist simultaneously multiple modes of antibacterial action in ZnO. Accepting proposed models of NPs internalization [42,43] to explain the reported correlation between the size and cytotoxicity, one should take into consideration the enhancement of other modes such as surface-surface interactions in our microparticle solutions. By using inversion during incubation, we ensure a consistent interaction between crystalline free surfaces and bacteria by increasing the frequency of such interactions, hence improving the overall cytotoxicity. Such an increase may not be observed at the nanoscale as FE-SEM images in Figure 6a show that ZnO NPs do not exhibit the significant, large-scale surface degradation observed in ZnO MPs (Figure 6b). This could be attributed to differential effects of particle solubility.

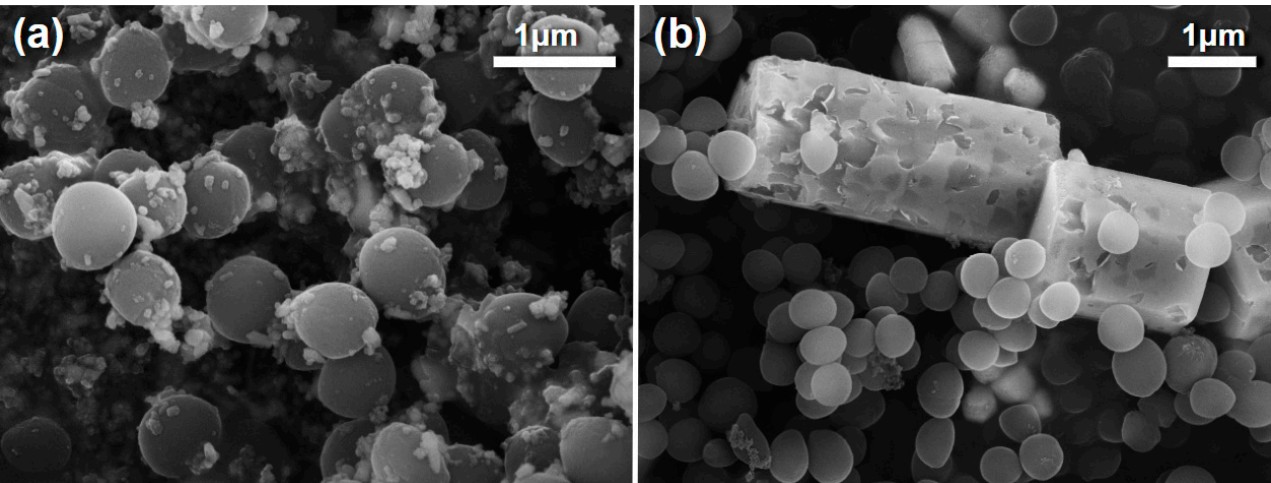

**Figure 6.** FE-SEM images comparing the differences in bacterial interaction of *S. aureus* with the free crystalline surface of differing scales of ZnO in MHB where (**a**) shows interaction with commercial ZnO NPs and (**b**) shows interactions with hydrothermally grown ZnO MPs [29].

The aforementioned surface degradation in MPs demonstrates that they undergo only partial dissolution of the crystalline surfaces. NPs, on the other hand, experience more complete solubility due to the much smaller volume of available bulk and hence they do not exhibit a notable increase in surface area. Additionally, these SEM images indicate adsorption of ZnO NPs to the bacterial surface not observed in MP samples. This behavior may further limit availability of surface-surface interactions as NPs occupy interaction sites of the bacterial surfaces thus restricting areas available for potential interaction with other surfaces or uptake of potentially cytotoxic ions and compounds.

Besides and importantly, in addition to differences in ZnO particle responses to bacterial environments we observe formation of secondary crystalline phases in solutions after such interactions. These new crystals, depicted in Figure 7, are orders of magnitude larger than the initial ZnO particles and exhibit pyramidal morphologies regardless of that of the ZnO precursors.

The secondary crystalline phase crystals are found in all biological environments utilized in this study except the saline solution without bacteria. It should be noted that only a very small concentration of such crystals was observed in saline environments containing *S. aureus*. We find that the quantity of these crystals is increasing with the concentration of aqueous phosphates. The secondary crystalline phase is the most prevalent in PBS containing *S. aureus*. This lines up well with the EDXS results shown in Figure 8. Here, one finds significant concentrations of Zn, O and P with residual carbon from the tape utilized in mounting for SEM. Obviously, phosphorus is not originally present in saline media in any appreciable quantity. The morphology and chemical composition of these pyramidal crystals are indicative of monoclinic zinc phosphides [44–46] though further structural studies are needed to confirm their exact nature. Observation of such crystalline phases has been reported previously for ZnO NPs within metal–organic frameworks [47]. In those studies, the phosphorus-containing phase was classified as zinc phosphate hydrate, was present at negligible abundances and thus was not the focus of the discussion about its effect on antibacterial action. Moreover, this phase was present in a form of a relatively random assortment of nanoscale morphologies as opposed to the pyramidal microparticles observed herein.

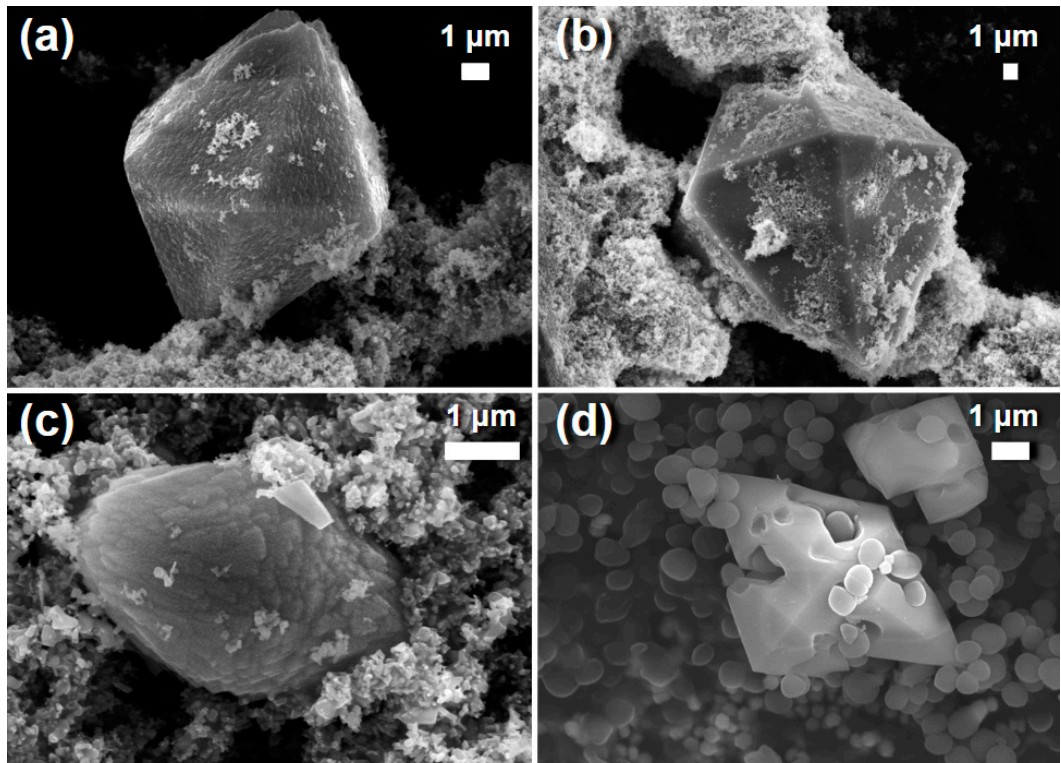

**Figure 7.** FE-SEM images of secondary crystalline phases in solutions with: (**a**) commercial ZnO NPs after exposure to PBS in isolation; (**b**) commercial ZnO NPs after exposure to PBS with *S. aureus*; (**c**) commercial ZnO NPs after exposure to saline with *S. aureus*; (**d**) hydrothermally grown ZnO MPs after exposure to MHB with *S. aureus*.

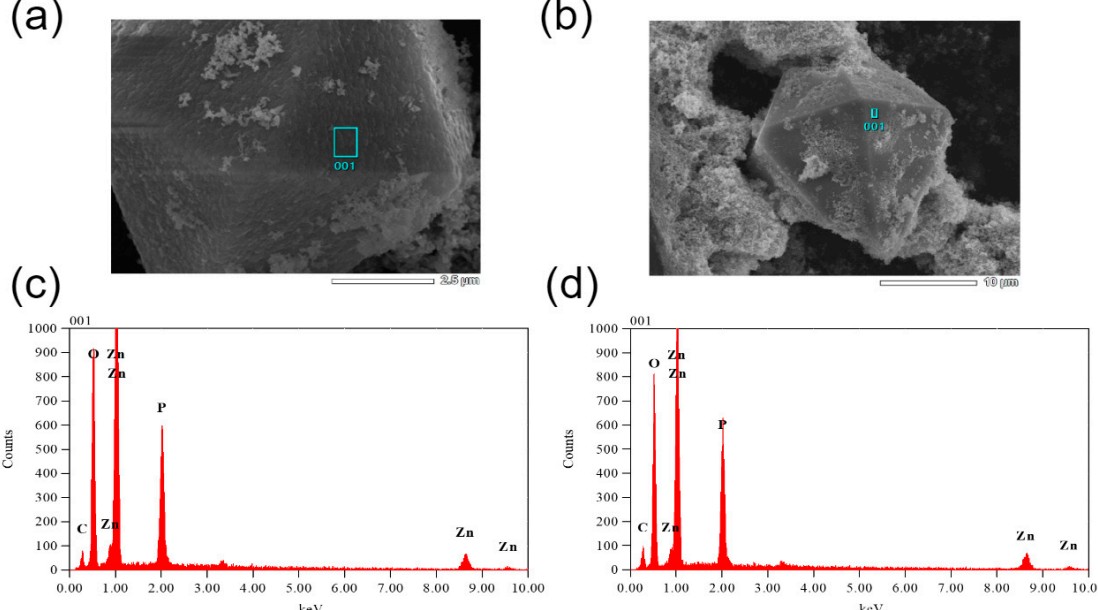

**Figure 8.** EDXS analyses of secondary crystalline phases following exposure to biological environments. SEM images indicate the EDXS scan area, in blue, on the observed secondary crystalline phase resulting from commercial ZnO NPs after exposure to (**a**) PBS and (**b**) PBS containing *S. aureus* bacteria. The resulting EDXS spectra are shown for commercial ZnO NPs after exposure to (**c**) PBS and (**d**) PBS containing *S. aureus* bacteria. For both spectra the reported atomic concentrations are as follows: Zn $8 \pm 1\%$, O $64 \pm 1\%$, P $7 \pm 1\%$, C $21 \pm 1\%$.

Formation of these crystals can be attributed to two different processes. In one of them, ZnO particles serve as nucleation sites for development of the secondary larger structures. Alternatively, partial dissolution of ZnO particles results in excess $Zn^{2+}$ ions which then react with the aqueous phosphorus compounds and crystallize. Considering the composition and structure of the secondary phase crystals in addition to the solubility of ZnO observed in biological environments (Figure 6), an argument for the latter explanation can be made. Formation of these larger crystals in bacteria-free environments further demonstrates that these crystals are the result of media-specific interactions. PBS alone produces significant concentrations of this phase without bacteria to mediate any interactions with the ZnO particles. That is not to say that bacteria are irrelevant in this process. They do produce secondary phase crystals in saline solution although not in any meaningful quantity. This may be the result of bacteria-bacteria interactions or ZnO-bacteria interactions. In low phosphate environments *S. aureus* has been shown to produce teichoic acid degrading enzymes thus leading to leaching of phosphate compounds from the bacterial cell wall of other staphylococci [48]. These compounds may potentially interact with free $Zn^{2+}$ ions and crystallize. Additionally, interactions with ZnO particles or their products result in disruption of the cellular membranes releasing phosphorus compounds into solution for crystallization to occur. Differences in the surface quality of this secondary phase are seen in the presence of bacteria. This may indicate differences in the phosphate compounds present in the media compared to those freed from the cell wall. To that, a slight increase in the concentration of aqueous phosphorus compounds due to bacteria leads to more favorable growth conditions. More detailed studies into the nature and growth mechanics of these crystals are required to make a definite determination.

The development of this crystalline phase is significant in the context of the antibacterial mechanisms of ZnO as it is associated with a significant decrease in bacterial growth inhibition (Figure 9). One can observe a decrease in the bacterial growth inhibition over the range of two orders of magnitude in PBS solutions where these secondary crystals are abundant as opposed to the case of saline which is comparatively sparsely populated with the secondary phase. Inclusion of controls for bacterial growth in media without ZnO particles demonstrates that this is not a byproduct of media interactions with bacteria but rather an interplay between media, ZnO and bacteria.

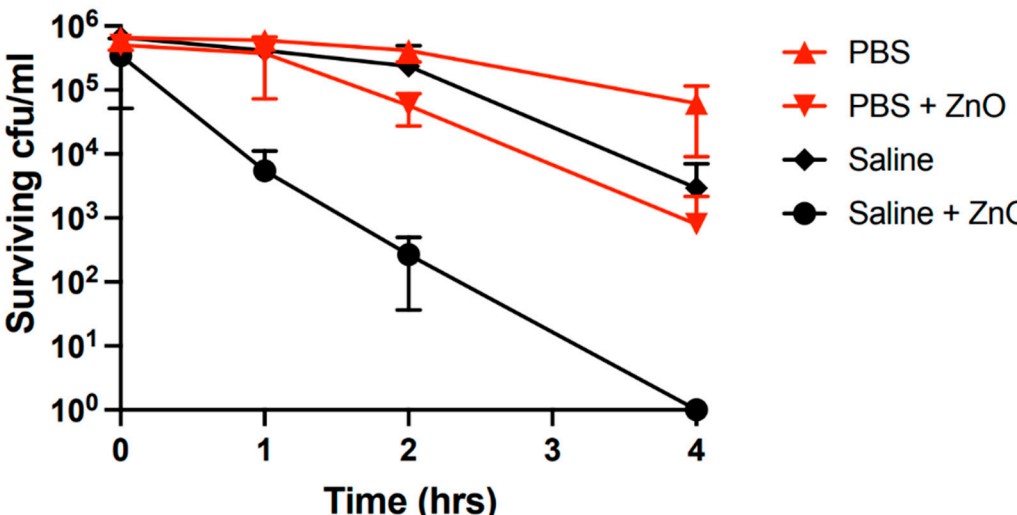

**Figure 9.** Survival of *S. aureus* bacteria over time in different growth environments with and without the presence of ZnO NPs.

We suggest that there exist competitive processes between ZnO, bacteria and growth media. The competition for "resources" toxic to bacteria or changes in the interaction intensity between bacteria and ZnO caused by the presence of aqueous phosphates, or the development of the secondary phase would result in a decrease in bacterial growth inhibition. Aqueous phosphates may occupy surface trap states hence restricting them from serving as interaction sites with bacteria. Alternatively, formation of the phosphorus-containing crystals may reduce free ion concentration of $Zn^{2+}$ in solution thus reducing their availability for toxic uptake of these ions by *S. aureus*. Furthermore, the secondary phase crystals themselves may serve as interaction sites for bacteria. In Figure 7d we see a significant interaction of bacteria with these crystals creating extended surface defects that may propagate deeper into the bulk. Such interactions replace potential harmful interactions with ZnO thus increasing survivability of the bacterial population.

Further evidence of such competitive interactions is seen in the room temperature PL spectra of ZnO particles after exposure to these biological environments. At both the nano- and microscale we observe significant shifts in the NBE peak luminescence energy. For ZnO NPs (Figure 10) we observe a redshift in the NBE emission at RT for both saline and PBS environments with and without the presence of *S. aureus*. Spectra for saline environments exhibit a minor shift of ~25 meV with little to no difference observed in the presence of *S. aureus*. For the case of PBS, a larger shift of ~100 meV in the peak energy is observed and the presence of bacteria has a more significant impact. In this environment, the addition of *S. aureus* reduces the observed shift to only ~50 meV.

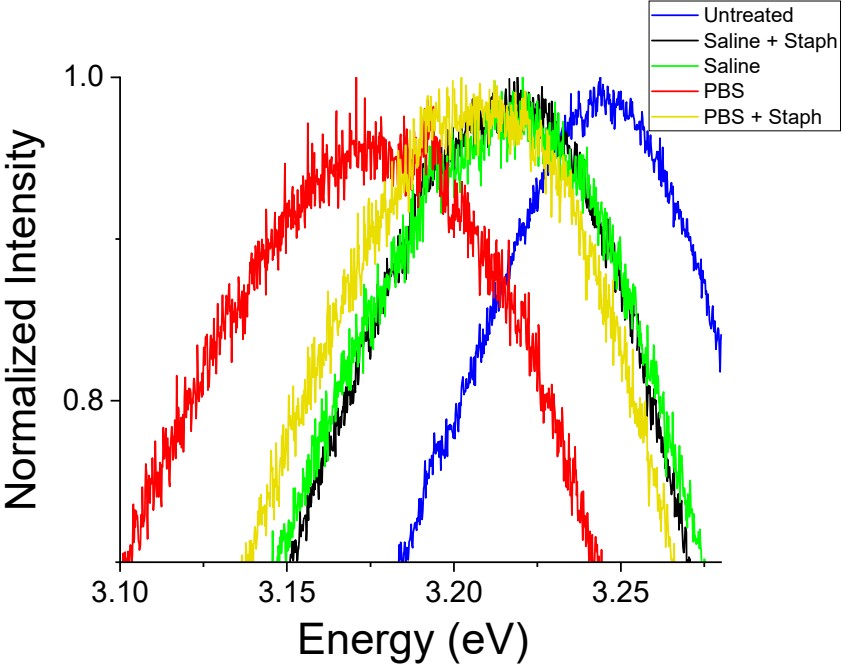

**Figure 10.** Room temperature PL spectra of the NBE region normalized to the NBE peak intensity for commercial ZnO NPs after exposure to biological environments.

On the other hand, for ZnO MPs there is a blue shift in the peak energy in response to exposure to the environments studied (Figure 11). This difference in directionality of the observed shift is not necessarily indicative of differences in the nature of their interactions but more likely a size effect as the extended surface defects observed in Figure 6 would serve to increase the free surface area in ZnO MPs whereas the more complete dissolution of NPs reduces the free surface area with accompanying changes in their respective band gap energies.

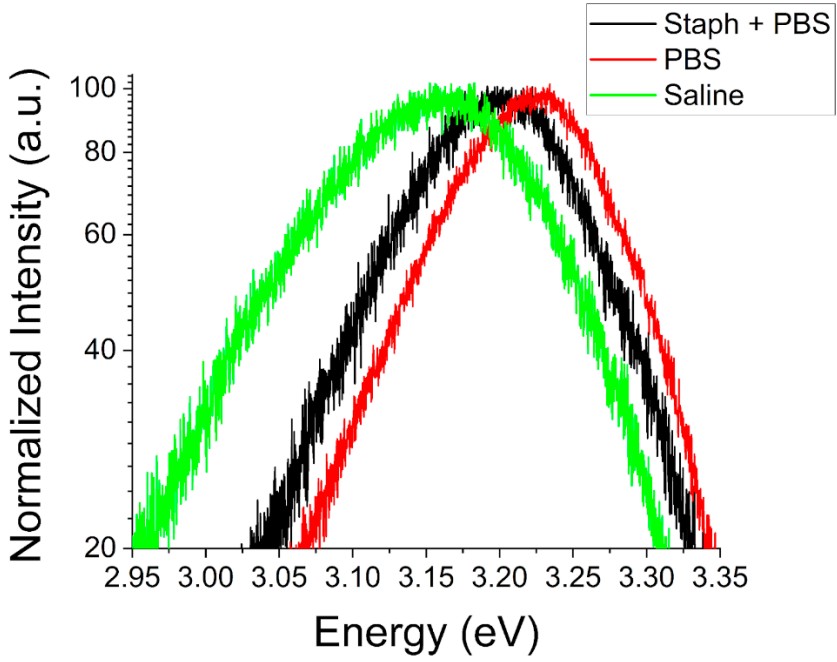

**Figure 11.** Room temperature PL spectra of the NBE region normalized to the NBE peak intensity for a type B ZnO MP sample after exposure to *S. aureus* in PBS as well as PBS and saline media in isolation.

This blue shift may also be attributed to adsorption of aqueous phosphate molecules and complexes to surface trap states. As discussed previously, this would result in the passivation of these states and would influence the surface band bending. Despite this change, the effects of bacteria would remain the same as in the case of NPs where a negligible effect is observed in saline environments and a notable reduction of this shift in PBS. This suggests identical interactions of bacteria, either with ZnO or with those media components responsible for the observed shift in PBS. It is known that *S. aureus* requires phosphates for certain metabolic processes [49,50]. Therefore, bacteria may actively desorb the adsorbed phosphorus-containing compounds from the ZnO surfaces. It can be concluded that bacteria may be interacting with and thereby freeing the surface traps occupied by aqueous phosphates or merely consuming compounds involved in their passivation. This model supports the idea of competitive interactions between bacteria and PBS with ZnO.

One possible route of this competition may be contest for the free $Zn^{2+}$ ions. Previous studies addressed the role of growth media on $Zn^{2+}$ production as a means to determine the overall solubility of ZnO NPs [51–53]. It was reported that in PBS there exist negligible free ion concentrations, despite the introduction of ZnO NPs. The authors concluded that ZnO is relatively insoluble in such environments or that aqueous phosphates in solution either preferentially bind to ZnO surfaces that would ordinarily act as a source of free ion release or that such compounds are highly reactive with $Zn^{2+}$ ions [53]. The clearly observed surface degradation of our MPs (Figure 6), in addition to the observed development of the secondary crystalline phase (Figure 7) rules out the notion that ZnO is insoluble in phosphate-rich environments. Our results indicate that aqueous phosphates both react significantly with free $Zn^{2+}$ ions and bind to surface trap states which would ordinarily act as sources of free ion release.

In Figure 12 we show room temperature PL spectra of the deep defect region after exposure to bacterial growth environments. One can observe that the intensity of the deep defect luminescence is modified following the exposure to biological environments. There is a significant increase of the overall intensity of the deep defect luminescence in all exposed samples with those exposed to saline exhibiting the greatest luminescence and to

PBS—the lowest. An increase in deep defect luminescence is consistent with the extended degradation of the free crystalline surfaces shown in Figure 6.

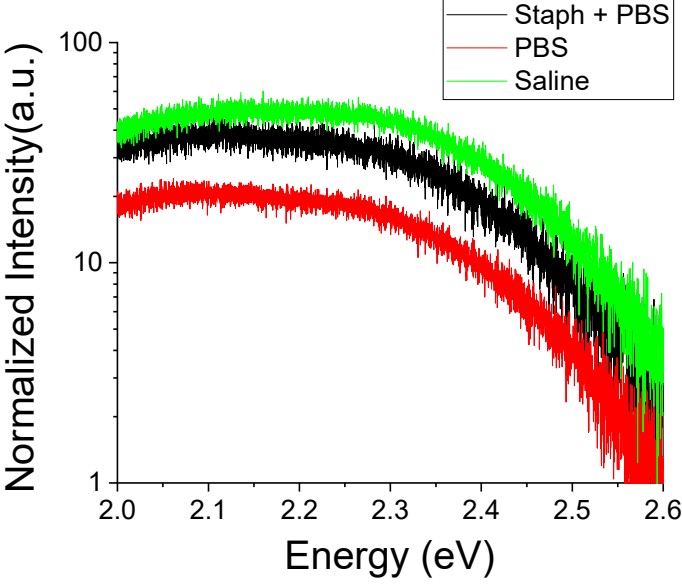

**Figure 12.** Room temperature PL spectra of deep defect luminescence regions normalized to the noise floor for a type B ZnO MP sample at 300 K after exposure to *S. aureus* in PBS as well as PBS and saline media in isolation.

The presence of *S. aureus* bacteria again counteracts the effects of PBS by enhancing the luminescence in samples exposed to bacterial growth environments while having a negligible impact in saline. The nature of the deep defect luminescence is also modified. Prior to the exposure there was an imbalance in peaks attributed to Zn and O deficiencies. O deficiency was rather pronounced in the PL spectra, especially in samples with more polar surfaces (Figure 4). These morphology differences may impact their potential for bacterial growth inhibition. An abundance of surface trap states and presence of oxygen deficiencies at the surface would lead to an excess of Zn atoms at polar surfaces which can be released and taken in by *S. aureus*. This can be mediated by the solubility of the crystalline surface in the ambient or a direct surface contact with bacteria. After exposure we see in all environments a reduction of the relative intensity of the peak at ~2.46 eV associated with O deficiency. This is indicative of media interactions resulting in the dissolution of the free crystalline surface releasing free $Zn^{2+}$ ions into solution. The decrease in deep defect luminescence in PBS may point to a decreased solubility of ZnO in this environment or, more likely, a stabilization of the crystalline structure via surface defect occupation by aqueous phosphates. Such occupation of surface defect trap states occurs likely at the expense of bacterial interactions since in the presence of bacteria the deep defect luminescence increases (i.e., increased interaction with the surface states).

Therefore, the source of competition between the phosphate-rich growth media and *S. aureus* may center around the direct interaction with the surface trap states instead of, or in addition to, the free $Zn^{2+}$ ion consumption. Earlier in this paper (cf. discussion of Figure 2) we considered discrepancies in the excitonic spectral structure of as-received ZnO MPs vs. NPs and suggested influence of different abundances of surface trap states. When comparing the low temperature excitonic spectra for ZnO NPs before and after exposure to *S. aureus* bacteria in MHB one can see a significant suppression of luminescence in this spectral region after exposure (Figure 13). This difference is most prominent in the surface BEx region ~3.368 eV (surface defect bound excitons) followed by the FEx emission ~ 3.378 eV with a more minor effect on the bulk defect bound excitonic peaks. The suppression in the intensity of the surface BEx emission may indicate adsorption of various compounds onto the surface defect traps or removal of these sites by bacteria.

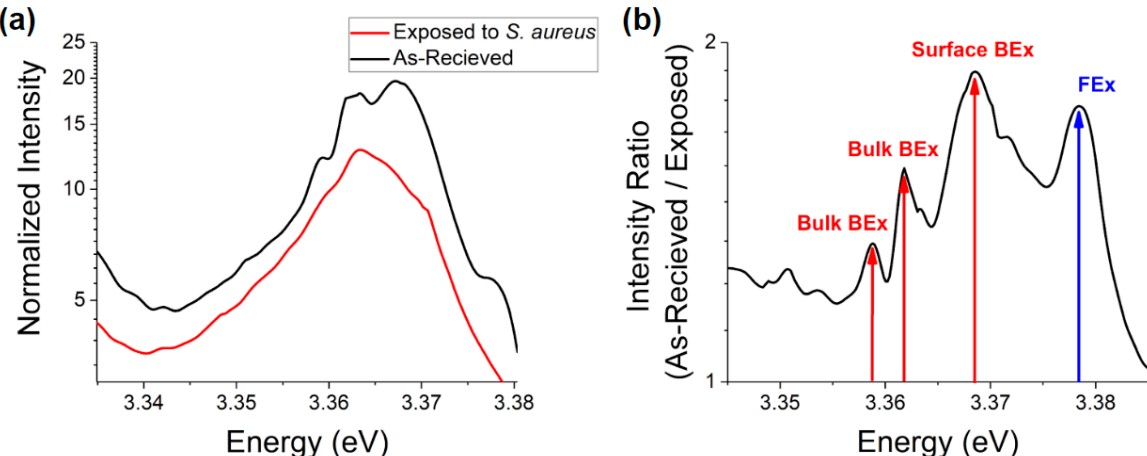

**Figure 13.** 10 K PL spectra of: (**a**) excitonic recombination for commercial ZnO NPs normalized to the noise floor before and after exposure to *S. aureus* bacteria in MHB growth media and (**b**) the ratio of the spectra shown in part (**a**).

When considering the large shift in ZnO NP NBE luminescence in phosphate rich media and the suppression of this shift in the presence of *S. aureus* in conjunction with the decrease of antibacterial efficacy in these environments one can conclude that interaction with the surface trap states is a highly competitive process between both phosphate rich media and *S. aureus* bacteria. Occupation of defect sites by aqueous phosphates would prevent their interaction with bacteria and thus the abundance of surface trap states may contribute to bacterial growth inhibition. If so, this mechanism would be enhanced at the microscale, supporting our hypothesis that the similarities in cytotoxicity can be explained by different competing mechanisms with relative dominance depending on the particle scale.

We investigated the effects of bacterial exposure on the optoelectronic response of ZnO MPs with different morphologies. In Figure 14 showing the NBE PL spectra for type LR and HP samples before and after exposure to *S. aureus* in MHB, it can be seen that regardless of the predominant surface type a blue shift in the NBE peak energy is observed. The crystals with more polar surfaces are less influenced by this interaction (Figure 14a) in comparison to their less polar counterparts (Figure 14b). This may occur due to an increase in the surface charge carriers or due to occupation of surface trap states at the non-polar surfaces. In the first case, the more polar samples exhibit a higher initial excess of surface and near-surface charge thus saturating much quicker whereas the non-polar surfaces can experience a larger shift before reaching this point. In this case, interactions with the bacterial growth media result in a charge transfer to the ZnO free surface either from the bulk or the bacterial cell wall. In the second case, the more non-polar surfaces, initially less saturated with oxygen deficiency (cf. Figure 4a), will be more susceptible to develop additional oxygen vacancy and zinc interstitial defects, leading to greater shifts of the NBE peaks (Figure 14b). On the other hand, the more polar surfaces, with initially greater amount of such defects, will gain less of oxygen deficiency leading to smaller shifts of the NBE peaks (Figure 14a). The generation of such surface defects may be the result of the particle solubility due to the media interactions discussed above. Thus more sites for bacterial interaction could be created leading to a comparable growth inhibition observed for our MPs in previous studies [29]. We suggest that in the absence of the growth media effects, the antibacterial interactions occur at higher rates on polar surfaces where the surface trap states are of greater abundance. Such impact of surface polarity could explain the driving mechanisms behind the observed bacterial growth inhibition and merits further investigation.

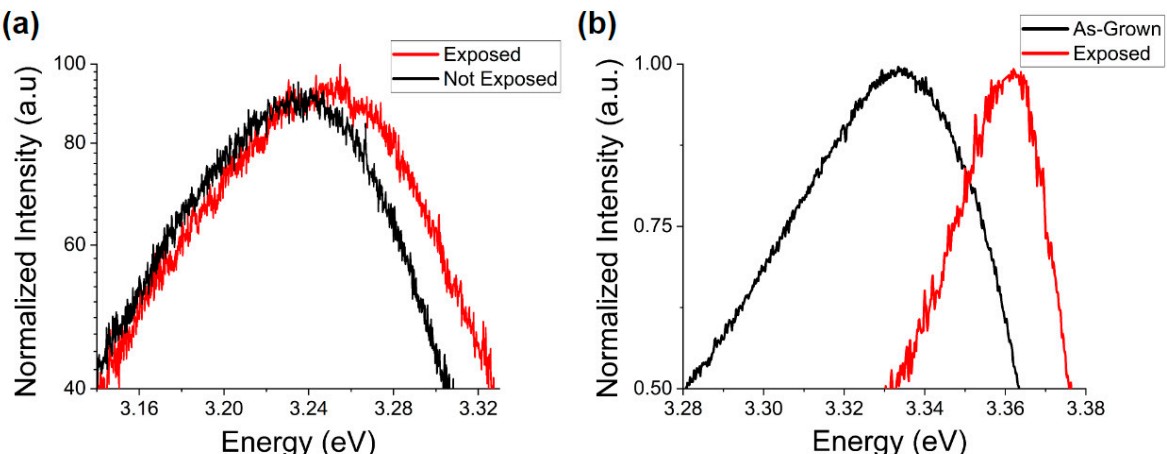

**Figure 14.** Room temperature PL spectra of the NBE region normalized to the NBE peak intensity for ZnO MPs before and after exposure to *S. aureus* in MHB: (**a**) type HP sample and (**b**) type LR sample.

## 4. Conclusions

ZnO particles at the nano- and microscale demonstrate potential to serve as critical components of antibacterial coatings and agents capable of combatting both wild type and traditionally antibiotic-resistant strains of bacteria. As such, the findings herein provide insight as to conditions of both the media and free crystalline surfaces that could maximize performance in these applications. In this work we studied commercial ZnO NPs and hydrothermally in-house grown MPs before and after exposure to PBS, saline and MHB bacterial growth media with or without *S. aureus* bacteria. These hydrothermally grown MPs exhibit a greater abundance of surface defects than the commercial NPs. Differences in the NBE and deep defect regions of PL spectra for differing morphologies of ZnO MPs point to the polar surfaces being more oxygen deficient than non-polar surfaces. Despite these differences, MIC antibacterial assay results show comparable levels of cytotoxicity across scales and morphologies. Based on these observations we suggest that multiple competing mechanisms of antibacterial action in ZnO may be at play. Moreover, particle sizes and/or morphologies may affect which of the inhibitory mechanisms dominate. This hypothesis is supported by differences in the post-assay surface degradation observed with FE-SEM and discrepancies in the spectral signatures of optically active defects probed with PL. Our results demonstrate significant defect-mediated growth media-bacteria-ZnO interactions at all scales. Additionally, we find a significant development of a secondary phosphorus-containing crystalline phase during these interactions. Formation of this phase strongly correlates with a decreased bacterial growth inhibition. This fact emphasizes the role of the growth media, particularly those containing aqueous phosphates, in both the solubility of ZnO and its interactions with bacteria. We propose that competitive processes exist between aqueous phosphates and *S. aureus* bacteria affecting driving mechanisms of antibacterial action. In particular, changes observed for optically active defects points to production of excess $Zn^{2+}$. This, in turn, may occur either due to the direct interaction of *S. aureus* with surface defects or the increased solubility of associated surfaces leading to excess $Zn^{2+}$ in the environment.

**Author Contributions:** Conceptualization, J.M.R., Y.M.S. and S.M.M.; methodology, J.M.R.; validation, Y.M.S. and S.M.M.; formal analysis, D.J.; investigation, D.J., A.C., J.M.R. and I.A.; resources, Y.M.S. and S.M.M.; data curation, I.T. and D.J.; writing—original draft preparation, D.J.; writing—review and editing, Y.M.S.; visualization, D.J.; supervision, Y.M.S. and S.M.M.; project administration, Y.M.S. All authors have read and agreed to the published version of the manuscript.

**Funding:** This work was supported in part by the National Science Foundation, Grant No. PHY-1852267.

**Institutional Review Board Statement:** Not applicable.

**Informed Consent Statement:** Not applicable.

**Data Availability Statement:** Not applicable.

**Acknowledgments:** The authors would like to thank Iman Ali and Samuel Garcia-Rodriguez for their efforts and the following high-school students participating in the TCU Research Apprentices Program for technical assistance: P. Ahluwalia, V. Athipatla, S. Bidare, M. Hattarki, R. Jadeja, R. Maheshwari, R. Northern, L. Rogers, A. Speights, and Y. Zaidi.

**Conflicts of Interest:** The authors declare no conflict of interest.

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
