# Peer review of "Influence of Surface Properties and Microbial Growth Media on Antibacterial Action of ZnO"

_coatings, doi:10.3390/coatings12111648_

Round 1

Reviewer 1 Report

Bacterial infections pose a growing threat to global health and food security as antibiotic-resistant strains become increasingly common and difficult to treat.

The problems associated with the treatment of such resistant strains using traditional antibiotics are caused by powerful adaptive reactions of bacteria in the form of genetic mutations and lateral gene transfer. This makes the development of new antibiotics increasingly difficult and less profitable, which necessitates the search for unconventional antibacterial agents for use in sanitation and medicines.

Such agents should have reduced toxicity, increased selectivity and effectiveness against existing strains resistant to antibiotics.

A wide range of ZnO compounds demonstrates a noticeable antibacterial effect against antibiotic-resistant strains of Gram-positive and Gram-negative bacteria under various conditions with selective toxicity and effectiveness in combating biofilm formation. ZnO is a common inexpensive material with significant photocatalytic efficiency and high biocompatibility, recognized by the U.S. Food and Drug Administration as a safe substance for external use.

In this study for commercial ZnO nano-powders and homegrown ZnO microparticles of varying morphologies the response to bacterial growth media in isolation and with Staphylococcus aureus bacteria was investigated. In this study, ZnO microparticles are synthesized by a controlled hydrothermal method and subjected to biological analysis in various microbial environments. Changes in the optoelectronic, structural and chemical properties of these crystals before and after such exposure are characterized using temperature-dependent photoluminescence spectroscopy, scanning electron microscopy and energy-dispersive X-ray spectroscopy. Various bacterial media were used in the studies to determine the effect of the interaction of media on the cytotoxic efficacy of ZnO. Studies have shown that the nature of structural and optoelectronic changes in ZnO depends both on the type of medium and on the presence (or absence) of bacteria in these media. Based on these observations, the authors suggest that numerous competing mechanisms of the antibacterial action of ZnO may be involved. Moreover, particle size and/or morphology can influence which of the inhibitory mechanisms dominates.

The work undoubtedly deserves to be published in the journal Coatings.

Author Response

We thank the reviewer for their efforts in reviewing this manuscript.

Reviewer 2 Report

Reviewer's comments:

In this work, the morphologies of commercial ZnO nano-powders and home grown ZnO microparticles were investigated, the response to bacterial growth media in isolation and with Staphylococcus aureus bacteria were also discussed. Changes in the optoelectronic, structural and chemical properties of these crystals before and after such exposure are characterized utilizing temperature-dependent photoluminescence spectroscopy, scanning electron microscopy and energy-dispersive X-ray spectroscopy. Although this paper presents some new results, some minor revisions should be made before publication.  

1. Please give the important conclusion in the Introduction.

2. For page 15 and line 24, Zinc oxide is widely used in food storage and biomedicine. In addition, they are also widely used in metallurgy, chemical industry, materials and other fields [1,2]. I suggest the author cite the following latest report. Improving oxidation resistance of TZM alloy by deposited Si-MoSi2 composite coating with high silicon concentration; Microstructure evolution and growth mechanism of Si-MoSi2 composite coatings on TZM (Mo-0.5Ti-0.1Zr-0.02 C) alloy.

3. Some large size rod like ZnO were observed in Fig. 6 (b), which is obviously different from Fig. 6 (a). Please give a detailed explanation.

4. I suggest using Figures (a), (b), (c) and (d) to divide Fig. 8. Figs. 8 (a) and (b) should use the same magnification. The element concentrations of O, Zn and P should also be given on Figs. 8 (a) and (b).

Author Response

Point 1: 1. Please give the important conclusion in the Introduction.

Response 1: We are grateful to the reviewer for the recommendation to state important concluions in the introduction. We have accordingly, added mention of some of our key conclusions on p.4 lines 152-159. It now reads “These results described below, are indicative of the existence of competitive processes be-tween aqueous phosphates in the bacterial growth environment and S. aureus bacteria that influence the driving mechanisms of antibacterial action. We demonstrate interactions with optically-active defects associated with the production of excess Zn2+. The changes resulting from these interactions are shown to occur due to either direct interactions of S. aureus with ZnO surface oxygen deficiencies or the increased solubility of associated sur-faces leading to excess Zn2+ in the bacterial environment.”

Point 2: For page 15 and line 24, Zinc oxide is widely used in food storage and biomedicine. In addition, they are also widely used in metallurgy, chemical industry, materials and other fields [1,2]. I suggest the author cite the following latest report. Improving oxidation resistance of TZM alloy by deposited Si-MoSi2 composite coating with high silicon concentration; Microstructure evolution and growth mechanism of Si-MoSi2 composite coatings on TZM (Mo-0.5Ti-0.1Zr-0.02 C) alloy.

Response 2: 
We could not find such quote on page 15, line 24, which reads “The crystals with more polar surfaces are less influenced by this interaction (Figure 14a) in comparison to their less polar counterparts (Figure 14b).” However, on p.2 lines 64 - 66 we say in our manuscript: “Additionally, ZnO is a stable compound of high durability and heat resistance leading to widespread usage in antibacterial coatings for food storage [3,4] and biomedicine [9,10,12,14].”. If this is the intended passage, we have therefore added on p.2. line 66 “, metallurgy, and chemical industries”, as requested. The reference mentioned by the reviewer is not dealing with Zn or ZnO whatsoever, so we chose not to add it to the list of cited references.

Point 3: Some large size rod like ZnO were observed in Fig. 6 (b), which is obviously different from Fig. 6 (a). Please give a detailed explanation.

Response 3: The reviewer is correct. The microscale rods shown in Fig. 6(b), after interaction with bacteria, are very similar to those shown in Fig. 1(c), before interaction with bacteria. On the other hand, nanoparticles shown in Fig. 6(a), after interaction with bacteria, are very similar to those shown in Fig. 1(a), before interaction with bacteria. We hope this explanation makes the distinction clear for the referee. We have therfore revised the caption for Fig. 6 appropriately to make this distinction apparent. It now reads “FE-SEM images comparing the differences in bacterial interaction of S. aureus with the free crys-talline surface of differing scales of ZnO in MHB where a) shows interaction with commercial ZnO NPs and b) shows interactions with hydrothermally grown ZnO MPs [29].”

Point 4: I suggest using Figures (a), (b), (c) and (d) to divide Fig. 8. Figs. 8 (a) and (b) should use the same magnification. The element concentrations of O, Zn and P should also be given on Figs. 8 (a) and (b).

Response 4: We thank the referee for the valuable suggestion. We have separated the figure into four parts as recommended, and revised the captions. Unfortunately, we do not have SEM images for the EDXS collection with the same magnification, however the scale bars and the EDXS-analyzed areas on both are clearly indicated. We have also incorporated the suggestion and included elemental concentrations were added into the captions. It now reads “EDXS analyses of secondary crystalline phases following exposure to biological environments. SEM images indicate the EDXS scan area on the observed secondary crystalline phase resulting from commercial ZnO NPs after exposure to a) PBS and b) PBS containing S. aureus bacteria. The resulting EDXS spectra are shown for commercial ZnO NPs after exposure to c) PBS and d) PBS containing S. aureus bacteria. For both spectra the reported atomic concentrations are as follows: Zn 8 ± 1%, O 64 ± 1%, P 7 ± 1%, C 21 ± 1%.”

Reviewer 3 Report

I have reviewed the manuscript coatings-1975295 entitled "Influence of Surface Properties and Microbial Growth Media on Antibacterial Action of ZnO" proposed by Dustin Johnson et al. for publication in the Coatings. The results are interesting. However, some parts of the manuscript should be improved before it is considered to be acceptable. In my opinion, the following points have to be considered:

1. The authors did not explain the novelty and significance of their work in the introduction section. Moreover, this section is not cohesive. Indeed, this section is intended to "convey the core findings of the paper", i.e. reflect the best novelty of this paper in a concise form. The authors shall show the best novelty of the work, such as how your research advances the state-of-the-art of the topic/area, and /or how much better is your work compared with peer researchers on the same or similar topics. At the end of this section, the main objective of this study must be mentioned.

2. On page 2, line 64, the sentence “Additionally, ZnO is a stable compound of high durability and heat resistance leading to widespread usage in antibacterial coatings for 65 food storage and biomedicine.” please also add the anti-corrosion properties of ZnO that have been mentioned in the literature: Surface and Coatings Technology 360 (2019): 153-171, Materials Letters 258 (2020): 126779.

3.  The captions to the figures must give full details so that the information can be clearly understood by the reader, for example in Fig. 8.

4.  The quality of all figures needs to be improved (fig.13a). Please remember that these figures will be shrunk once the paper is published and no one can read anything as a result.

5.  The English of the whole paper is good, but some errors could still be found. Therefore, the English of the paper should be reviewed.

6.  The conclusion is too long. Please extend that. Also, in the conclusion, part, specify the application of this work clearly, and also provide the application of this particular material with key features in one or two sentences.

Author Response

Point 1: The authors did not explain the novelty and significance of their work in the introduction section. Moreover, this section is not cohesive. Indeed, this section is intended to "convey the core findings of the paper", i.e. reflect the best novelty of this paper in a concise form. The authors shall show the best novelty of the work, such as how your research advances the state-of-the-art of the topic/area, and /or how much better is your work compared with peer researchers on the same or similar topics. At the end of this section, the main objective of this study must be mentioned.

Response 1: We gratefully appreciate this suggestion. The corresponding revisions were added on p.3, lines 128 - 136. It now reads “This is a novel approach since previous studies in the field concentrated primarily on the response of bacteria or the corresponding bacterial environments. In our work, we inves-tigate the influence of the ZnO free surfaces in antibacterial interactions and changes therein. These investigations differentiate themselves from the current landscape of re-search in this area as we utilize microscale particles to both eliminate internalization and to better elucidate the interactions at the free crystalline surface with a specific polarity. We focus on not just the influence of growth media on cytotoxicity but on isolating those in-teractions between just ZnO and growth media as well as those with bacteria.”

Point 2: On page 2, line 64, the sentence “Additionally, ZnO is a stable compound of high durability and heat resistance leading to widespread usage in antibacterial coatings for 65 food storage and biomedicine.” please also add the anti-corrosion properties of ZnO that have been mentioned in the literature: Surface and Coatings Technology 360 (2019): 153-171, Materials Letters 258 (2020): 126779.

Response 2: We added on p.2. line 66 “, metallurgy, and chemical industries”. The appropriate references mentioned by the reviewer were added.

Point 3: The captions to the figures must give full details so that the information can be clearly understood by the reader, for example in Fig. 8.

Response 3: We thank the reviewer for this suggestion and have therefore revised the captions for Figure 8 in order to more clearly convey that they are the EDXS spectra of the phosphate based crystals resulting from exposure to PBS environments with and without the presence of s.aureus bacteria. We have also included associated elemental concentrations in the caption. It now reads “EDXS analyses of secondary crystalline phases following exposure to biological environments. SEM images indicate the EDXS scan area on the observed secondary crystalline phase resulting from commercial ZnO NPs after exposure to a) PBS and b) PBS containing S. aureus bacteria. The resulting EDXS spectra are shown for commercial ZnO NPs after exposure to c) PBS and d) PBS containing S. aureus bacteria. For both spectra the reported atomic concentrations are as follows: Zn 8 ± 1%, O 64 ± 1%, P 7 ± 1%, C 21 ± 1%.”

Point 4: The quality of all figures needs to be improved (fig.13a). Please remember that these figures will be shrunk once the paper is published and no one can read anything as a result.

Response 4: We appreciate the feedback and the attention to detail the reviewer provides. Figure 13 has been updated for visibility as have Figures 1, 2, 3, 4, 6, 7, 8, 10, 11, 12 and 14.

Point 5: The English of the whole paper is good, but some errors could still be found. Therefore, the English of the paper should be reviewed.

Response 5: The manuscript has been carefully reviewed and approved by a native English speaker. If the referee can suggest any specific changes, the authors will happily consider them.

Point 6: The conclusion is too long. Please extend that. Also, in the conclusion, part, specify the application of this work clearly, and also provide the application of this particular material with key features in one or two sentences.

Response 6: Since the first two sentences of this remark contradict each other, we assume that the referee meant that the conclusion is too short based on the reviewers suggestion to include an additional “one or two sentences”. Please correct us if we are mistaken. Otherwise, following the referee’s input, we added specific mentions of potential applications of our results, on p.16 lines 579 - 583. It now reads “ZnO particles at the nano- and microscale demonstrate potential to serve as critical components of antibacterial coatings and agents capable of combatting both wild type and traditionally antibiotic-resistant strains of bacteria. As such, the findings herein pro-vide insight as to conditions of both the media and free crystalline surfaces that could maximize performance in these applications.”

Reviewer 4 Report

Dear Authors,

I have included all comments in the attachment.

Best Regards

Author Response

Point 1: Abbreviations, formulae, units: conform to acceptable standards.

Lines e.g. 190, 193, 195: the unit should be “mL”. Please correct in all the text.

Response 1: We thank the reviewer for pointing out this error and inconsistency in the notation. We have updated “ml” to the appropriate “mL” throughought section “2.3. Antimicrobial Assays”

Point 2: Lines e.g. 14, 26, 44, 53, 66, 83, 92: THERE ARE DOUBLE SPACES. Please correct the double spaces throughout the text !!

Response 2: We appreciate this suggestion. As per the reviewer’s request, all double spaces in the lines listed above in addition to those at the beginning of every sentence have been replaced with a standard single space.
